# Modulation of Dietary Choline Uptake in a Mouse Model of Acid Sphingomyelinase Deficiency

**DOI:** 10.3390/ijms24119756

**Published:** 2023-06-05

**Authors:** Ángel Gaudioso, Pilar Moreno-Huguet, Josefina Casas, Edward H. Schuchman, María Dolores Ledesma

**Affiliations:** 1Centro Biologia Molecular Severo Ochoa (CSIC-UAM), 28049 Madrid, Spain; pilar.morenoh@estudiante.uam.es; 2RUBAM, IQAC-CSIC & CIBEREHD, 08034 Barcelona, Spain; fina.casas@iqac.csic.es; 3Icahn School of Medicine at Mount Sinai, New York, NY 10029, USA; edward.schuchman@mssm.edu

**Keywords:** choline, sphingomyelin, lysosomal storage disorder, lipidomic, acid sphingomyelinase deficiency, phospholipid

## Abstract

Acid sphingomyelinase deficiency (ASMD) is a lysosomal storage disorder caused by mutations in the gene-encoding acid sphingomyelinase (ASM). ASMD impacts peripheral organs in all patients, including the liver and spleen. The infantile and chronic neurovisceral forms of the disease also lead to neuroinflammation and neurodegeneration for which there is no effective treatment. Cellular accumulation of sphingomyelin (SM) is a pathological hallmark in all tissues. SM is the only sphingolipid comprised of a phosphocholine group linked to ceramide. Choline is an essential nutrient that must be obtained from the diet and its deficiency promotes fatty liver disease in a process dependent on ASM activity. We thus hypothesized that choline deprivation could reduce SM production and have beneficial effects in ASMD. Using acid sphingomyelinase knock-out (ASMko) mice, which mimic neurovisceral ASMD, we have assessed the safety of a choline-free diet and its effects on liver and brain pathological features such as altered sphingolipid and glycerophospholipid composition, inflammation and neurodegeneration. We found that the choline-free diet was safe in our experimental conditions and reduced activation of macrophages and microglia in the liver and brain, respectively. However, there was no significant impact on sphingolipid levels and neurodegeneration was not prevented, arguing against the potential of this nutritional strategy to assist in the management of neurovisceral ASMD patients.

## 1. Introduction

Acid sphingomyelinase deficiency (ASMD) is a lysosomal storage disorder caused by mutations in the gene-encoding acid sphingomyelinase (ASM) [1]. Besides peripheral organ impact, including hepatomegaly and splenomegaly [2], the infantile and chronic neurovisceral forms of ASMD also exhibit neuroinflammation and neurodegeneration [3]. Although the peripheral symptoms of the disease can be addressed by intravenous infusion of the recombinant enzyme [4], the brain pathology remains unaddressed, triggering the search for new therapeutic approaches [5].

The biochemical hallmark of ASMD is the abnormal accumulation of sphingomyelin (SM) in different cells and tissues, including the Central Nervous System (CNS) [6]. High SM levels lead to lysosomal permeabilization and autophagy impairment [7] in all ASM deficient cells, as well as calcium imbalance [8] and synaptic alterations [9,10,11] in ASM deficient neurons. Therefore, reducing SM levels is a main therapeutic target in ASMD. SM is comprised of a polar head group, phosphocholine, and a hydrophobic backbone, ceramide, and is the only sphingolipid with a phosphocholine group in its structure [12]. The direct action of sphingomyelinases, both acid and neutral, cleaves the phosphocholine group transforming the SM into ceramide [13]. Phosphocholine is a derivative of choline, an essential nutrient that must be obtained from the diet [14]. Choline is necessary for the synthesis of acetylcholine [15], methyl donors such as betaine or S-adenosylmethionine [16] and choline-containing lipids: phosphatidylcholine, lysophosphatidylcholine, choline-containing ether lipids and SM [17].

Choline deficient diets, including choline-free and methionine-choline free diets, induce liver steatosis and lipid deposition in the liver that promotes fatty liver disease in mouse models, and therefore have been used to model Non-Alcoholic Fatty Liver Disease (NAFLD) [18]. It has been reported that ASM activity plays a role in the development of NAFLD-related liver damage caused by these diets [19]. Notably, however, in the context of a choline-methionine deficient diet, the lack of ASM confers resistance against hepatic endoplasmic reticulum stress. Moreover, ASM inhibition protects wild type (wt) mice against hepatic steatosis, fibrosis and liver damage under this diet [20]. These findings suggested a reduced risk of affecting the liver in ASMko mice treated with a choline-free diet, and that such a diet might contribute to reduce SM accumulation by limiting its production. We have herein assessed this hypothesis by treating ASMko mice with a choline-free diet for eight weeks. The results showed that this diet was not toxic to ASMko mice during this time of treatment and did not induce liver damage. Among the effects of the diet in the liver, we observed changes in the levels of certain sphingolipid and glycerophospholipid species and decreased macrophage size. Choline deprivation also reduced microglia size and therefore activation in the brain, but had minor impact in the lipid composition and did not prevent neurodegeneration.

## 2. Results

### 2.1. Choline-Free Diet Shows No Overt Toxicity and Does Not Affect Liver to Body Weight Ratio in ASMko Mice

We started to feed wt and ASMko mice a control or choline-free diet at eight weeks of age. At this stage, Purkinje cell degeneration in the anterior, but not posterior, lobes of the cerebellum is already evident, causing motor impairment among the first disease symptoms in the ASMko mice [21]. The choline-free diet was extended for eight weeks, a period in which Purkinje cell loss and SM accumulation progress in untreated ASMko mice [21,22]. No overt adverse effects (hair loss, diarrhoea or premature death) were observed in any of the mouse groups receiving the treatment. Weight gain was monitored once per week. All the experimental groups showed a similar weight gain during the first five weeks of treatment. In the last three weeks, the ASMko mice fed the choline-free diet did not gain weight (Figure 1A). Consistently, we observed a reduction in the daily food consumption (1.2-fold less) in these mice compared to the ASMko mice fed with the control diet (Figure 1B). Wt mice fed with a choline-free diet did not stop gaining weight and their food consumption was not significantly altered (Figure 1A,B). Next, we analyzed if choline-deprivation had effects on the liver size, since choline-methionine deficient diets had been shown to increase the liver to body weight ratio [23]. Control-diet fed ASMko mice had a higher (1.3-fold) liver to body weight ratio than wt mice, consistent with the characteristic hepatomegaly associated with ASMD [2]. Choline deprivation had no effects on this ratio. Liver weight was also not affected by choline deprivation in the wt mice (Figure 1C).

### 2.2. Choline-Free Diet Alters Lipid Composition in the Liver, Mainly by Changing Levels of Sphingolipids and Glycerophospholipids with 34:1 Fatty Acids

Because of the important role of choline for sphingolipid and glycerophospholipid metabolism and the reported anomalies in the liver of NAFLD mouse models, we studied the effects on these classes of lipids of choline-deprivation in this organ. Levels of more than seventy species from eight different lipid classes [sphingolipids: LysoSM (Lysosphingomyelin); SM; dhSM (dihydrosphingomyelin); Cer (ceramide); dhCer (dihydroceramide); glycerophospholipids: PC (phosphatidylcholine); PE (phosphatidylethanolamine); PS (phosphatidylserine)] were quantified by liquid chromatography/high resolution mass spectrometry (LC/HRMS).

To determine if the wt and ASMko mice could be segregated according to their sphingolipidomic and/or glycerophospholipidomic composition in the liver, we performed a Partial Least Square Discriminant Analysis (PLS-DA) of the data. As expected, ASM deficiency changed the overall lipidomic composition in the liver compared to wt (Figure 2A,B). Choline deprivation further segregated the lipidomic composition in both mouse genotypes (Figure 2A,B). Analysis of total SM levels confirmed the drastic increase (35.6-fold) in control diet-fed ASMko mice compared to wt [22] (Figure 2C). Choline deprivation reduced SM levels in the wt mice (1.3-fold reduction) but did not significantly change the SM accumulation in the ASMko mice (Figure 2C). SM comprises a heterogenous group of species differing in their fatty acid length and degree of saturation. Different levels of accumulation and toxicity depending on the organ has been recently reported for specific SM species in the ASMko mice [24]. We thus analyzed the levels of the different SM species in the liver of control- and choline-free diet fed ASMko and wt mice. The levels of all SM species increased in the ASMko mice compared to wt (Figure 2D). The choline-free diet did not change the levels of any of the species in the ASMko mice, but reduced SM species with longer chain fatty acids (22–24 carbons) in the wt mice (22:0: 1.4-fold; 22:1: 2.2-fold; 24:0: 1.3-fold; 24:1: 1.6-fold; 24:2: 1.4-fold) (Figure 2D). This SM species-specific effect of the choline-free diet in the wt mice altered the relative abundance of the individual SM species. By increasing the relative abundance of short SM species (16:0 and 18:0) and decreasing the longer ones (24:1), the choline-free diet rendered the SM species profile in wt mice similar to that in ASMko mice, although the overall abundance of SM remained much lower in the wt mice (Figure 2E). Choline deprivation did not affect the levels of other sphingolipids (LysoSM, dhSM, Cer and dhCer) in wt and ASMko mice (Appendix A).

Besides SM, PC is another major lipid class that contains choline moieties. We therefore analyzed total PC levels in liver extracts and, although not statistically significant, we detected an increase in ASMko mice compared to wt (1.2-fold increase) (Figure 2F). Analysis of PC species also revealed slight effects of the choline-free diet in both ASMko and wt mice in which the levels of shorter and less unsaturated PC species decreased (34:1: 1.3-fold in wt and 1.1-fold in ASMko; 36:2: 1.1-fold in wt and 1.2-fold in ASMko), while species with longer fatty acids and a higher number of unsaturated bonds increased (38:4: 1.1-fold in wt and 1.1-fold in ASMko; 38:6: 1.3-fold in wt and 1.3-fold in ASMko; 40:6: 1.4-fold in wt and 1.2-fold in ASMko) (Figure 2G). These changes did not have a major impact on the relative abundance of PC species (Figure 2H).

Under choline-deficient conditions, the phosphatidylethanolamine methyl transferase (PEMT) can produce PC by addition of methyl groups to PE [25]. This could explain the reduction in PE levels we observed in wt (1.1-fold) and ASMko (1.3-fold) mice fed with choline-free diet compared to control diet (Figure 2I). As observed for PC, the PE most reduced was the 34:1 species, both in wt and ASMko mice (1.5- and 1.6-fold reduction, respectively) (Figure 2J). Additionally, there were two PE species, 36:2 and 38:4, that showed significant higher levels in ASMko compared to wt mice fed with control diet (36:2: 1.4-fold increase in ASMko compared to wt; 38:4: 2.6-fold increase in ASMko compared to wt) and were reduced by choline retrieval (36:2: 1.4 fold decrease; 38:4: 2.3-fold decrease) (Figure 2J). Regarding the relative abundance of PE species, the only noticeable change observed was that of 34:1 PE, which was slightly decreased by 3% in wt and ASMko mice (Figure 2K).

### 2.3. Choline-Free Diet Reduces Inflammation in the Liver of ASMko Mice

Inflammation in the liver characterizes all forms of ASMD and can be monitored by the number and size of macrophages. In order to analyze if the choline-free diet induced liver inflammation or if, on the contrary, it prevented the pathological inflammation that is present in the ASMko mice, we analyzed macrophages by immunocytochemical staining using antibodies against the specific protein marker F4/80. This analysis confirmed the inflammation in the liver of ASMko mice, which showed increased macrophage number (1.5-fold) and size (3.2-fold) compared to wt mice (Figure 3A,B). The choline-free diet reduced macrophage size (1.2-fold) but not the number in the ASMko mice, supporting an anti-inflammatory effect (Figure 3A,B). This diet did not affect macrophage number or size in the wt mice, pointing to the absence of pro-inflammatory effects of the choline retrieval in these mice, at least under our experimental conditions.

### 2.4. Choline Deficiency Had Minor Effects on Brain Lipid Composition

Severe brain pathology characterizes the infantile neurovisceral ASMD and the ASMko mice. In the ASMko mice, the cerebellum is the most affected brain area showing early death of Purkinje cells, astrocytosis and microglia activation [26]. We thus analyzed the effects of the choline-free diet in this brain area. LC/HRMS analysis detected 104 different species from the 8 lipid classes already mentioned (sphingolipids: LysoSM; SM; dhSM; Cer; dhCer; glycerophospholipids: PC; PE; PS).

PLS-DA analysis of the data indicated the clear distinction of the cerebellar sphingolipid and glycerophospholipid content between wt and ASMko mice fed with control diet (Figure 4A,B). Choline deprivation had minor effects on these differences (Figure 4A,B).

Levels of SM, LysoSM and dhSM were significantly increased (2.4-, 4.1- and 3.6- fold, respectively) in the cerebellum of ASMko compared to wt mice fed with control diet, while the levels of Cer and dhCer were not changed (Figure 4C and Appendix A). The choline-free diet did not have effects on the levels of any of these lipids in the wt and ASMko cerebellum (Figure 4C and Appendix A). Analysis of SM species showed increased levels of all species in the ASMko compared to wt cerebellum. Choline deprivation did not alter SM species’ levels or their relative abundance in any of the mouse genotypes (Figure 4D,E).

Levels of PC were also significantly increased (1.3-fold) in the cerebellum of ASMko compared to wt mice fed with control diet (Figure 4F). Choline-free diet did not revert this increase (Figure 4F). Changes in PC species did not seem to depend on the length or saturation of the fatty acids, since we observed increments in the levels of PC 32:0, 34:1 and 38:6 (Figure 4G). The only significant effect promoted by the choline-free diet was the slight decrease (1.1-fold) of PC 34:1 in the wt mice (Figure 4G). No significant changes in the relative abundance of PC species were observed in any of the genotypes (Figure 4H).

PE levels were slightly (1.5-fold), although not significantly, increased in the cerebellum of ASMko compared to wt mice fed with control diet (Figure 4I). When PE species were individually analyzed, the increment reached significance in the ASMko mice for PE 36:2 (1.8-fold), PE 38:4 (1.6-fold) and PE 38:5 (2.1-fold) (Figure 4J). The only significant effect promoted by the choline-free diet was the reduction (1.5-fold) of PE 34:1 species in the ASMko mice (Figure 4J). The relative abundance of PE species was similar between control and choline-free diet (Figure 4K).

### 2.5. Choline-Free Diet Reduces Neuroinflammation but Not Cell Death in the Brain of ASMko Mice

Purkinje cell death in the cerebellum was monitored by inmunocytochemical staining of the specific protein marker of these cells: Calbindin. A drastic reduction of Purkinje cells (3.9-fold) was evident in the ASMko compared to wt mice fed with control diet (Figure 5A,B). The choline-free diet did not prevent this pathological feature in the ASMko mice (Figure 5A,B). In agreement, the motor impairment observed in the ASMko mice in the rotarod test (they spent 2.0-fold less time in the rod than wt mice) was not ameliorated by choline deprivation (Figure 5C). The choline-free diet did not affect motor abilities in the wt mice (Figure 5B).

Inflammation in the cerebellum was monitored by immunocytochemical staining of the astrocytic marker GFAP and the microglia marker Iba-1. Increased GFAP intensity (3.4-fold) (Figure 5D,E) and greater number and size of microglia (2.9-fold and 3.8-fold, respectively) (Figure 5F,G) evidenced astrocytosis and microgliosis in the ASMko compared to wt mice fed with control diet. Choline-free diet did not affect GFAP intensity (Figure 5D,E), nor microglia number in the ASMko mice, but significantly reduced microglia size (1.4-fold) (Figure 5F,G), indicating a reduction in the activation of these cells. Choline deprivation did not have overt effects on astrocytes (Figure 5D,E) or microglia (Figure 5F,G) in the wt mice.

## 3. Discussion

Lipid alterations, inflammation and neurodegeneration characterize the infantile and chronic neurovisceral forms of ASMD. While the intravenous infusion of recombinant ASM is efficient to treat the peripheral disease [27], the inability of the enzyme to cross the brain blood barrier leaves brain pathology unaddressed. It is therefore important to find strategies suitable to treat the CNS disease manifestations in ASMD. Nutritional approaches are attractive since they facilitate non-invasive, long-term treatment in a pediatric population. The hypothesis of this study was that dietary deprivation of choline, an essential nutrient necessary for the production of SM, would reduce accumulation of this lipid, a key pathological event in all ASMD tissues.

Choline-restricted diets are commonly used for the study of liver diseases, mainly NAFLD [28]. In animal models, the most frequently used method is the methionine-choline deficient diet [29], which promotes profound changes in the hepatic lipidome of mouse models [30,31] and can be highly toxic if mantained for a long time [32,33]. A less toxic alternative is the choline-free diet that keeps the methionine content at a similar percentage (0.4%) as the control diet. This milder effect, together with pevious findings showing that ASM deficiency protected mice from developing fatty liver disease when being fed these diets [20], prompted us to assess a two month-long choline-free diet in ASMko mice, which mimic the neurovisceral phenotypes of ASMD. The results confirmed no overt toxicity of the diet, which did not alter liver weight, under the experimental conditions used. However, the reduction in weight gain observed in the last three weeks of treatment in the ASMko mice, which was accompanied by less food consumption, raise concerns about the effects of longer treatment. In addition, the results obtained showing lack of effects of the choline-free diet on SM and PC total levels failed to support our initial hypothesis. The complex and highly connected metabolic routes for sphingolipid and phospholipid synthesis and degradation (Figure 6) bring the opportunity to activate compensatory mechanisms that counteract choline deprivation.

Lipidomic analysis in the liver revealed that the choline-restricted diet had unexpected species-specific effects in the levels of sphingolipids and glycerophospholipids. Since species-specificity has not been described for the different isoforms of sphingomyelin synthases, we propose that these differential changes are due to choline-free diet induced alterations in the activity of enzymes that act earlier in the SM synthesis pathway, such as members of the Ceramide synthase (CerS) family [34]. The liver has a specific profile of CerS expression with high abundance of CerS2 [35] that is responsible for the synthesis of sphingolipids with very long fatty acids (20–26 carbons) [36], which are those reduced in wt mice fed with the choline-free diet. It could be that choline restriction diminishes CerS2 activity. Changes in long chain fatty acid sphingolipids were not observed in the choline-free fed ASMko mice. Compensatory alterations in the activity of enzymes contributing to the sphingolipid metabolic pathway, different from those occuring in wt mice, may account for this observation, and assessments of these enzymes is an interesting avenue for future investigation.

The hepatic changes observed in the levels of glycerophospholipids due to choline restriction were also species specific, with reduction of PC and PE species with shorter and less unsaturated fatty acids, and no significant changes in PS. Production of PC and PE in the liver can be achieved by the “de novo” synthesis through the so-called Kennedy pathway [37], or by the modification of preexisting phospholipids by the action of the enzymes Phosphatidylserine decarboxylase (PSD) and PEMT [38,39] (Figure 6). The Kennedy pathway depends totally on choline availability while PEMT activity, although depending also on choline levels, can operate if methionine is mantained in the diet [40] (Figure 6). PEMT activity is esentially restricted to hepatocytes [41]. Under prolonged choline-deprivation PEMT expression increases [25] and is the only source of PC [42].

The question therefore arises as to whether the changes observed in this work could be due to the modulation of the Kennedy pathway and/or of PSD, PEMT activities. It has been described that PEMT does not exhibit substrate specificity and metabolizes PE according to the mole percent of that lipid [43], and that PE species with mono- or di-unsaturated fatty acids are preferentially generated by the action of the Kennedy pathway rather than PSD activity [44,45]. These species are the ones that decrease their levels with the choline-free diet in wt and ASMko mice. Therefore, the downregulation of the Kennedy pathway is likely. On the other hand, the presence of methionine in the choline-free diet used in this study, and the consequent activity of PEMT to produce PC, could explain the failure to reduce SM levels we observed in the absence of dietary choline.

The lipidomic effects of choline deprivation in the brain were less evident than in the liver. The choline-free diet did not prevent the abnormal increase of SM, PC and of other lipid classes in the cerebellum of the ASMko mice. The activation of mechanisms to redistribute choline from different organs to the brain in choline-deficient states has been described [46]. They seem to respond to the need for enough choline in the brain to maintain its high phospholipid content [47], as well as the synthesis of the neurotransmitter acetylcholine [48]. While long-term, dietary restriction of choline can affect extra- and intracellular sources of substrates required for acetylcholine synthesis, and eventually limit acetylcholine release in the hippocampus [49], this may vary in discrete regions of the brain and compensatory mechanisms exist. Thus, prenatal choline deficiency leads to persistent upregulation of the expression of the high affinity choline transporter (CHT), which may serve to counteract the effect of low choline in the adulthood [50]. Moreover, levels of the acetylcholine-synthesizing enzyme choline acetyltransferase (ChAT) significantly increase in the brain of mice fed with choline-free diet [51]. Mechanisms of redistribution, and the action of PSD and PEMT, could explain why we did not see effects of the choline-free diet on the sphingolipid and glycerophospholipid composition of the brain. It might be that removal of methionine from diet, to also block PEMT-induced formation of phosphocholine, could be more efficient to reduce brain SM. Further experiments are required to confirm this hypothesis, and importantly to rule out that a more aggressive diet, such as the methionine-choline free diet, is not harmful.

The mild effects on the brain lipid composition of the choline-free diet may also explain the inability of the choline-free diet to prevent Purkinje cell death and motor impairment in the ASMko mice. We did observe the reduction of microglia size compatible with diminished inflammation. Choline deprivation also reduced macrophage size in the liver of ASMko mice. However, practical limitations of an extrapolation to human patients must be considered and longer-term studies are required to rule out possible toxic effects. It is likely that this treatment alone may not have a significant clinical impact.

## 4. Materials and Methods

### 4.1. Antibodies

Antibodies against the following proteins were used in immunohistochemical analysis: Calbindin (Swant, Austin, TX, USA; #300); F4/80 (Abcam, Waltham, MA, USA; #6640); GFAP (Merck, Rahway, NJ, USA; #MAB3402); Iba1 (Wako, Monza, Italy; #019-19741).

### 4.2. Mice

Breeding colonies were established from ASM heterozygous C57BL/6 mice [22]. Male and female ASMko and wt littermates were analyzed at 16 weeks of age. No gender-dependent differences were observed in any of the results. Procedures followed European Union guidelines and were approved by the CBMSO Animal Welfare Committee.

### 4.3. Choline-Free Diet Treatment

The choline-free diet was purchased from Envigo (#AIN-93G) and is a modification of the standard mice diet Teklad Global 18% Protein (Envigo, Indianapolis, IN, USA). Wt and ASMko mice were divided into two groups (*n* = 7 per group) receiving each of the two diets: Control Diet or Choline-Free Diet. Treatment started at eight weeks of age and the experiment was prolonged for another eight weeks.

### 4.4. Behavioural Tests

The rotarod test was performed in an accelerating rotarod apparatus (Ugo Basile, Varese, Italy), on which the mice were trained for two days at a constant speed: the first day—four times at four r.p.m. for one min; and on the second day—four times at eight r.p.m. for two min. On the third day, the rotarod was set to progressively accelerate from four to forty r.p.m. for five min, and the mice were tested four times. During the accelerating trials, the latency to fall from the rod was measured.

### 4.5. Liver Weight Assessment

Mice were weighed the day of sacrifice, and after perfusion the liver was extracted and weighed as well. Total liver weight was divided versus total mouse weight in order to determine the liver to body weight ratio.

### 4.6. Lipidomic Profiling

Lipids were analyzed as described [52,53], with minor modifications. In detail:

Phospholipids: A total of 750 µL of a methanol-chloroform (1:2, vol/vol) solution containing internal standards (16:0 D31_18:1 phosphocholine, 16:0 D31_18:1 phosphoethanolamine, 16:0 D31-18:1 phosphoserine, 0.2 nmol each, from Avanti Polar Lipids) were added to samples. Samples were vortexed and sonicated until they appeared dispersed and extracted at 48 °C overnight. The samples were then evaporated and transferred to 1.5 mL Eppendorf tubes after the addition of 0.5 mL of methanol. Samples were evaporated to dryness and stored at −80 °C until analysis. Before analysis, 150 µL of methanol were add to the samples, centrifuged at 13,000× *g* for 3 min, and 130 µL of the supernatants were transferred to ultra-performance liquid chromatography (UPLC) vials for injection and analysis.

Sphingolipids: A total of 750 µL of a methanol-chloroform (2:1, vol/vol) solution containing internal standards (N-dodecanoylsphingosine, N-dodecanoylglucosylsphingosine, C17-dihydrosphingosine and C17-dihydrosphingosine-1-phosphate, 0.2 nmol each, from Avanti Polar Lipids) were added to samples. Samples were extracted at 48 °C overnight and cooled. Then, 75 µL of 1 M KOH in methanol was added, and the mixture was incubated for 2 h at 37 °C. Following addition of 75 µL of 1 M acetic acid, samples were evaporated to dryness and stored at −80 °C until analysis. Before analysis, 150 µL of methanol were added to the samples, centrifuged at 13,000× *g* for 5 min and 130 µL of the supernatant were transferred to a new vial and injected.

Lipids were analyzed by liquid chromatography/high-resolution mass spectrometry (LC/HRMS). LC/HRMS analysis was performed using an Acquity ultra high-performance liquid chromatography (UHPLC) system (Waters, Milford, MA, USA) connected to a Time of Flight (LCT Premier XE) Detector. Full scan spectra from 50 to 1800 Da were acquired, and individual spectra were summed to produce data points each of 0.2 s. Mass accuracy at a resolving power of 10,000 and reproducibility were maintained by using an independent reference spray via the LockSpray interference. Lipid extracts were injected onto an Acquity UHPLC BEH C8 column (1.7 µm particle size, 100 mm × 2.1 mm, Waters, Ireland) at a flow rate of 0.3 mL/min and a column temperature of 30 °C. The mobile phases were methanol with 2 mM ammonium formate and 0.2% formic acid (A)/water with 2 mM ammonium formate and 0.2% formic acid (B). A linear gradient was programmed as follows: 0.0 min: 20% B; 3 min: 10% B; 6 min: 10% B; 15 min: 1% B; 18 min: 1% B; 20 min: 20% B; 22 min: 20% B.

Positive identification of compounds was based on the accurate mass measurement with an error <5 ppm and its LC retention time, compared with that of a standard (92%).

Quantification was carried out using the extracted ion chromatogram of each compound, using 50 mDa windows. The linear dynamic range was determined by injecting mixtures of internal and natural standards. Since standards for all identified lipids were not available, the amounts of lipids are given as nmol equivalents relative to each specific standard.

Glycerophospholipids (PC, phosphatidylcholine; PE, phosphatidylethanolamine; and PS, phosphatidylserine) were annotated using “total fatty acyl chain length: total number of unsaturated bonds” (e.g., 34:1). In the case of sphingolipids, (Cer, ceramide; dhCer, dihydroceramide; SM, sphingomyelin; dhSM, dihydrosphingomyelin; LysoSM, Lyso-sphingomyelin) as all the species have 18:1 fatty acid as the sphingoid base, they have only been denoted with the number of carbons and unsaturation of the other fatty acid (e.g., 16:0).

### 4.7. Immunohistochemistry

Mouse brains and livers were dissected, fixed in 4% PFA 0.12M sucrose, and cryoprotected for 24 h in 30% sucrose phosphate buffer saline. The tissue was then frozen in Tissue-Tek optimal cutting temperature compound (Sakura Finetek, Torrance, CA, USA), and 40-µm sagittal sections were obtained with a cryostat (CM 1950 Ag Protect freezing: Leica, Solms, Germany). The sections were incubated overnight at 4 °C with the primary antibodies and then with the corresponding Alexa-conjugated secondary antibodies. Finally, the sections were incubated for 10 min with DAPI (Merck), washed and mounted with ProLong Gold Antifade (Thermo Fisher, Waltham, MA, USA). Images were obtained on a confocal LSM710 microscope (Zeiss, Oberkochen, Germany) and quantified using the ImageJ software.

### 4.8. Statistical Analysis

Data from seven different animals per experimental group were quantified and presented as the mean ± SEM. Normality of the data was tested using the Shapiro–Wilk test. For multiple comparisons, data with a normal distribution were analyzed by one-way ANOVA followed by a Tukey post hoc test. The statistical significance of non-parametric data was determined by the Kruskal–Wallis test to analyze all experimental groups. *p*-values (*p*) < 0.05 were considered significant. In the figures, asterisks indicate the *p*-values: * *p* < 0.05; ** *p* < 0.01; *** *p* < 0.001; **** *p* < 0.0001. GraphPad Prism 6.0 software (GraphPad Software, La Jolla, CA, USA) was used for all statistical analysis.

The statistical analysis of the whole set of glycerophospholipids/sphingolipids from liver and cerebellum was performed using MetaboAnalyst 5.0 software [54]. The lipidomic data were normalized (mean-centered and divided by the standard deviation of each variable) and represented using a Partial Least Square Discriminant Analysis (PLS-DA) after autoscale of samples.

## Figures and Tables

**Figure 1 ijms-24-09756-f001:**
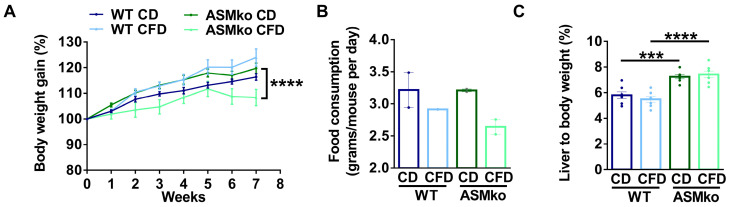
Choline-free diet effects on body and liver weight. (**A**) Weekly mean ± SEM body weight of wt and ASMko mice fed a control diet (CD) or a choline-free diet (CFD) (*n* = 7 mice per group; **** *p* < 0.0001 ASMko CD vs. ASMko CFD). (**B**) Daily food consumption (grams/mouse per day) of wt and ASMko mice fed a control diet or a choline-free diet (*n* = 2 cages per treatment). (**C**) Liver to body weight of wt and ASMko mice fed a control diet or a choline-free diet (*n* = 7 mice per group; *** *p* < 0.001; **** *p* < 0.0001).

**Figure 2 ijms-24-09756-f002:**
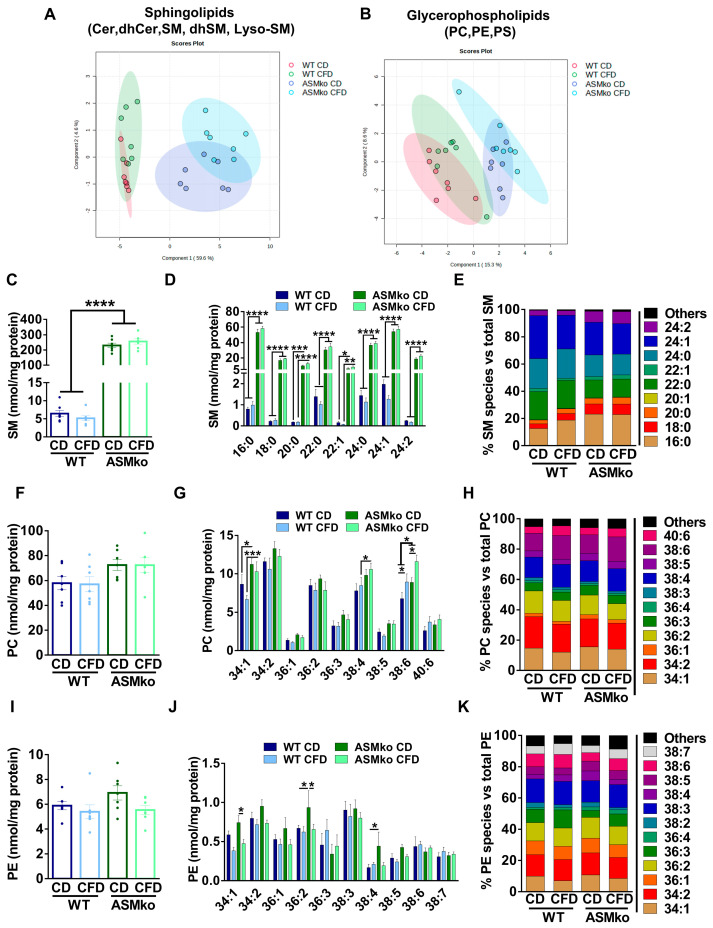
Choline-free diet alters the lipid composition of the liver in wt and ASMko mice. (**A**) Partial Least Square Discriminant Analysis of sphingolipidomics (Cer, dhCer, SM, dhSM, LysoSM) in liver of wt and ASMko mice fed a control diet or a choline-free diet (*n* = 7 mice per group). (**B**) Partial Least Square Discriminant Analysis of glycerophospholipidomics (PC, PE, PS) in liver of wt and ASMko mice fed a control diet or a choline-free diet (*n* = 7 mice per group). (**C**) Graphs show mean ± SEM of total SM levels expressed as nmol/mg protein in extracts from liver of wt and ASMko mice fed a control diet or a choline-free diet (*n* = 7; **** *p* < 0.0001). (**D**) Graphs show mean ± SEM of the indicated SM levels expressed as nmol/mg protein in extracts from liver of wt and ASMko mice fed a control diet or a choline-free diet (*n* = 7; * *p* < 0.05; ** *p* < 0.01; **** *p* < 0.0001). (**E**) Bars show mean percentages of the indicated SM species with respect to total SM in extracts from liver of wt and ASMko mice fed a control diet or a choline-free diet (*n* = 7 mice per group). (**F**) Graphs show mean ± SEM of total PC levels expressed as nmol/mg protein in extracts from liver of wt and ASMko mice fed a control diet or a choline-free diet (*n* = 7). (**G**) Graphs show mean ± SEM of the indicated PC levels expressed as nmol/mg protein in extracts from liver of wt and ASMko mice fed a control diet or a choline-free diet (*n* = 7; * *p* < 0.05; *** *p* < 0.001). (**H**) Bars show mean percentages of the indicated PC species with respect to total PC in extracts from liver of wt and ASMko mice fed a control diet or a choline-free diet (*n* = 7 mice per group). (**I**) Graphs show mean ± SEM of total PE levels expressed as nmol/mg protein in extracts from liver of wt and ASMko mice fed a control diet or a choline-free diet (*n* = 7). (**J**) Graphs show mean ± SEM of the indicated PE levels expressed as nmol/mg protein in extracts from liver of wt and ASMko mice fed a control diet or a choline-free diet (*n* = 7; * *p* < 0.05; ** *p* < 0.01). (**K**) Bars show mean percentages of the indicated PE species with respect to total PE in extracts from liver of wt and ASMko mice fed a control diet or a choline-free diet (*n* = 7 mice per group).

**Figure 3 ijms-24-09756-f003:**
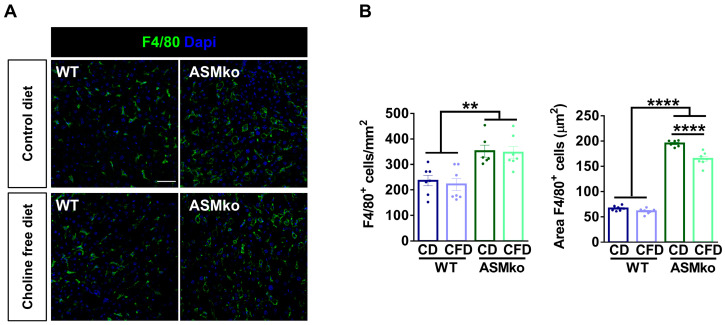
Choline-free diet reduces macrophage area in the liver of ASMko mice. (**A**) Immunohistochemical analysis against the macrophage marker F4/80 in the liver of wt and ASMko mice fed a control diet or a choline-free diet. DAPI staining shows cell nuclei. Bar = 50 µm. (**B**) Graph shows mean ± SEM number or area of F4/80+ cells (*n* = 7; ** *p* < 0.01; **** *p* < 0.0001).

**Figure 4 ijms-24-09756-f004:**
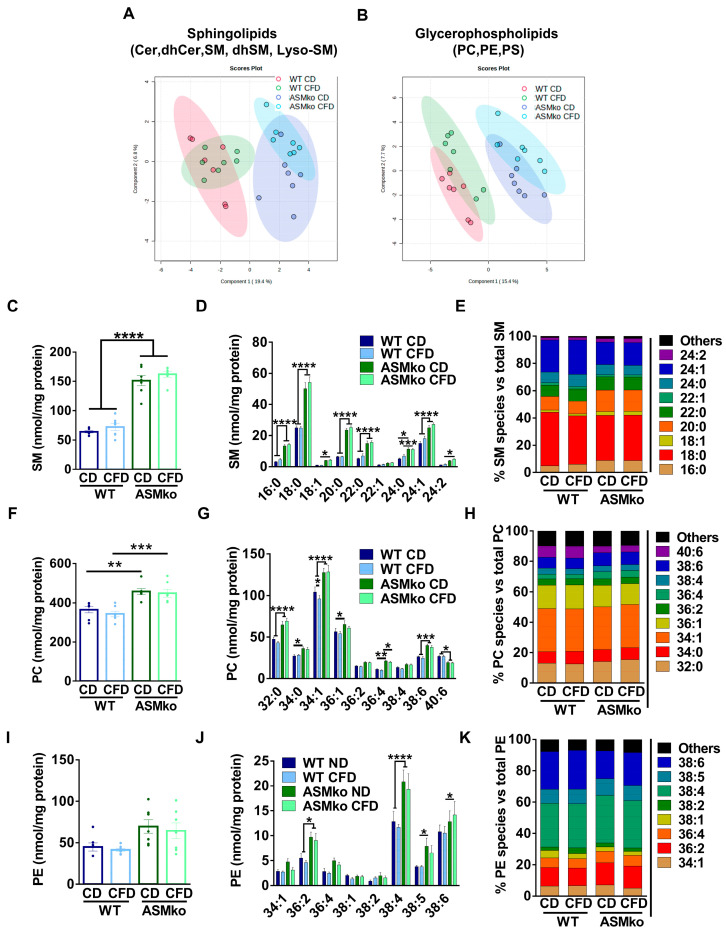
Choline deficiency had minor effects on brain lipid composition. (**A**) Partial Least Square Discriminant Analysis of sphingolipidomics (Cer, dhCer, SM, dhSM, LysoSM) in cerebellar extracts of wt and ASMko mice fed a control diet or a choline-free diet (*n* = 7 mice per group). (**B**) Partial Least Square Discriminant Analysis of glycerophospholipidomics (PC, PE, PS) in cerebellar extracts of wt and ASMko mice fed a control diet or a choline-free diet (*n* = 7 mice per group). (**C**) Graphs show mean ± SEM of total SM levels expressed as nmol/mg protein in extracts from cerebellum of wt and ASMko mice fed a control diet or a choline-free diet (*n* = 7; **** *p* < 0.0001). (**D**) Graphs show mean ± SEM of the indicated SM levels expressed as nmol/mg protein in extracts from cerebellum of wt and ASMko mice fed a control diet or a choline-free diet (*n* = 7; * *p* < 0.05; *** *p* < 0.001; **** *p* < 0.0001). (**E**) Bars show mean percentages of the indicated SM species with respect to total SM in extracts from cerebellum of wt and ASMko mice fed a control diet or a choline-free diet (*n* = 7 mice per group). (**F**) Graphs show mean ± SEM of total PC levels expressed as nmol/mg protein in extracts from cerebellum of wt and ASMko mice fed a control diet or a choline-free diet (*n* = 7; ** *p* < 0.01; *** *p* < 0.001). (**G**) Graphs show mean ± SEM of the indicated PC levels expressed as nmol/mg protein in extracts from cerebellum of wt and ASMko mice fed a control diet or a choline-free diet (*n* = 7; * *p* < 0.05; ** *p* < 0.01; *** *p* < 0.001; **** *p* < 0.0001). (**H**) Bars show mean percentages of the indicated PC species with respect to total PC in extracts from cerebellum of wt and ASMko mice fed a control diet or a choline-free diet (*n* = 7 mice per group) (**I**) Graphs show mean ± SEM of total PE levels expressed as nmol/mg protein in extracts from cerebellum of wt and ASMko mice fed a control diet or a choline-free diet (*n* = 7). (**J**) Graphs show mean ± SEM of the indicated PE levels expressed as nmol/mg protein in extracts from cerebellum of wt and ASMko mice fed a control diet or a choline-free diet (*n* = 7; * *p* < 0.05; **** *p* < 0.0001). (**K**) Bars show mean percentages of the indicated PE species with respect to total PE in extracts from cerebellum of wt and ASMko mice fed a control diet or a choline-free diet (*n* = 7 mice per group).

**Figure 5 ijms-24-09756-f005:**
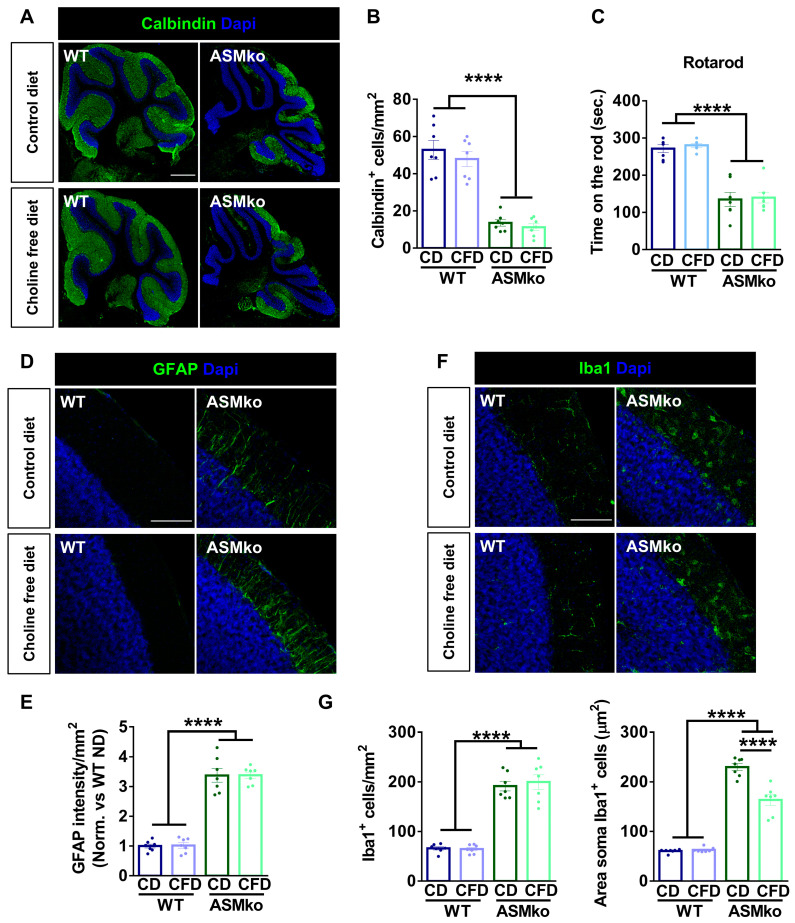
Choline-free diet did not prevent neurodegeneration but reduced microglia activation in the brain of ASMko mice. (**A**) Immunohistochemical analysis against the Purkinje cell marker Calbindin in the cerebellum of wt and ASMko mice fed a control or choline-free diet. DAPI staining shows cell nuclei. Bar = 500 µm. (**B**) Graph shows mean ± SEM number of Purkinje cells (*n* = 7; **** *p* < 0.0001). (**C**) Mean ± SEM time spent on the rod in the four trials of the rotarod test by wt and ASMko mice fed a control or a choline-free diet (*n* = 7 mice per group; **** *p* < 0.0001). (**D**) Immunohistochemical analysis against the astrocytic marker GFAP in the cerebellum of wt and ASMko mice fed a control diet or a choline-free diet. DAPI staining shows cell nuclei. Bar = 100 µm. (**E**) Graphs show mean ± SEM intensity associated to GFAP per area unit (*n* = 7; **** *p* < 0.0001). (**F**) Immunohistochemical analysis against the microglia marker Iba1 in the cerebellum of wt and ASMko mice fed a control diet or a choline-free diet. DAPI staining shows cell nuclei. Bar = 100 µm. (**G**) Graphs show mean ± SEM microglia number or area (*n* = 7; **** *p* < 0.0001).

**Figure 6 ijms-24-09756-f006:**
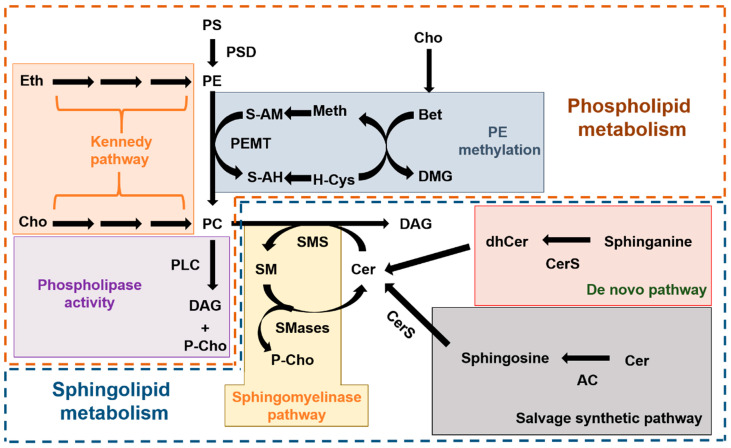
Metabolic routes of phospholipid and sphingolipid synthesis and degradation. Scheme summarizing the complex metabolic pathways for phospholipids (inside the brown dashed line) and sphingolipids (inside the blue dashed line) and their connection. In the phospholipid metabolism we highlight the “de novo” synthesis of PC and PE from choline and ethanolamine, respectively (also known as Kennedy pathway; orange box), the synthesis of PC due to PEMT-mediated methylation of PE (blue box) and the release of P-choline from PC mediated by Phospholipase C (PLD) (purple box). In the sphingolipid metabolism we highlight the three different pathways to produce Cer, which is the immediate precursor of SM: the sphingomyelinase pathway that releases P-choline (yellow box), the “de novo” pathway (red box) and the salvage pathway (grey box). AC: Acid ceramidase. Bet: Betaine. Cer: Ceramide. CerS: Ceramide synthase. Cho: Choline. DAG: diacylglycerol. dhCer: dihydroceramide. DMG: Dimethylglycine. Eth: ethanolamine. H-Cys: Homo-cysteine. Meth: Methionine. PC: phosphatidylcholine. P-Cho: Phospho-choline. PE: Phosphatidylethanolamine. PEMT: Phosphatidylethanolamine methyl transferase. PLC: Phospholipase C. PS: Phosphatidylserine. PSD: Phosphatidylserine decarboxylase. S-AH: S-adenosyl homocysteine. S-AM: S-adenosyl methionine. SM: Sphingomyelin. SMS: Sphingomyelin synthase. SMase: Sphingomyelinase.

## Data Availability

Not applicable.

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
