# Peer review of "Modulation of Dietary Choline Uptake in a Mouse Model of Acid Sphingomyelinase Deficiency"

_ijms, 2023, doi:10.3390/ijms24119756_

Round 1

Reviewer 1 Report

Gaudioso, A. et al reported the results on CD and CFD on wt and ASMko mice and proposed that choline restriction in diets may help ameliorate some of the defects resulted from acid sphingomyelinase deficiency. The effective reduction of SM accumulation by dietary choline restriction is expected to happen in both types of mice. However, there are some self-contradicting observations that seem to be at odds with the underlying molecular principles the authors wanted to advance.  The proposed molecular mechanisms lack a solid foundation that the mice of wt or ASMko genotypes are indeed responsive differently as proposed by the authors. Positive or negative controls agreeing with the proposed molecular mechanisms are not present. These flaws weakened the molecular basis for the proposed benefits of choline-deficient diets as a  treatment for ASM deficiency.

1. Given that choline needs to be taken from the diets as stated in the text, it is quite surprising that the syntheses of lyso-SM, dh-SM and SM are irresponsive to the choline-free diet (CFD) in Fig. S1 A, B, 2C,2D, in comparison with those treated with control diet in both wt and ASM-KO mice.

2. Similarly, the PC synthesis and global levels were not responsive to the CFD vs. CD either (Fig. 2F), despite some small differences in 38:4 and 38:6.

3. Analysis of shorter chain PCs (C18 or C20 acyl chains) that are more physiologically relevant was not done. The potential effects on Ach as a neurotransmitter were not analyzed either.

4. The same pattern was repeated in Figs. 3 and 4. The data do not support strong connections between CFD effects and the ASM knockout, because wt mice showed the same effects.

5. The data presented are difficult to understand at the metabolic circuit level, and need to have a clear physiological model for better presentation and easy understanding.

The presentation can be reorganized to help the readers understand the underlying metabolic effects. 

Author Response

REVIEWER 1

Gaudioso, A. et al reported the results on CD and CFD on wt and ASMko mice and proposed that choline restriction in diets may help ameliorate some of the defects resulted from acid sphingomyelinase deficiency. The effective reduction of SM accumulation by dietary choline restriction is expected to happen in both types of mice. However, there are some self-contradicting observations that seem to be at odds with the underlying molecular principles the authors wanted to advance.  The proposed molecular mechanisms lack a solid foundation that the mice of wt or ASMko genotypes are indeed responsive differently as proposed by the authors. Positive or negative controls agreeing with the proposed molecular mechanisms are not present. These flaws weakened the molecular basis for the proposed benefits of choline-deficient diets as a treatment for ASM deficiency.

We thank the reviewer for the criticisms and hope we have addressed them in this revision. We apologize if the original version was misleading regarding the hypothesis of our study. We did not propose that the molecular mechanisms causing lipidomic changes upon CFD would be different in the wt and ASMko mice. As the reviewer points out, the mild effects on lipidomics we have observed show a similar response to CFD in wt and ASMko mice, especially in the brain. What we postulated was that the lack of ASM, an enzyme that has been shown to mediate liver toxicity of methionine-choline free diets in wt mice, would avoid this toxicity in the ASMko mice. The results obtained confirm the lack of toxicity of CFD in the liver of ASMko mice and also show a safe profile in the wt mice at least in the experimental conditions used (2-month long treatment).

Answers to the specific queries:

  1. Given that choline needs to be taken from the diets as stated in the text, it is quite surprising that the syntheses of lyso-SM, dh-SM and SM are irresponsive to the choline-free diet (CFD) in Fig. S1 A, B, 2C,2D, in comparison with those treated with control diet in both wt and ASM-KO mice. 
  2. Similarly, the PC synthesis and global levels were not responsive to the CFD vs. CD either (Fig. 2F), despite some small differences in 38:4 and 38:6. 

We agree with this reviewer that the lack of significant effects of the CFD on the levels of lyso-SM, dh-SM, SM (query 1) and PC (query 2) is surprising. As a matter of fact, it is against our initial hypothesis that proposed this strategy to reduce sphingolipid levels in wt and ASMko mice. While unexpected, we believe this finding is an important message of our study that suggests the existence of compensatory mechanisms, including redistribution and the action of PSD and PEMT, to counteract choline deprivation.

  1. Analysis of shorter chain PCs (C18 or C20 acyl chains) that are more physiologically relevant was not done. The potential effects on Ach as a neurotransmitter were not analyzed either. 

The reason why we could not analyze C18 and C20 acyl chains in PCs is technical. Our analysis informs about the total number of carbons and unsaturations of the two fatty acids in the PC species but is unable to provide this information for the individual fatty acids. For instance, we cannot determine whether PC32:0 is composed by PC18:0/14:0 or 16:0/16:0. Different from PCs, this is not a limitation for the analysis of SM species since all of them contain a backbone sphingosine with 18:1 fatty acid allowing to infer the number of carbons and unsaturations of the other fatty acid.

Measuring Ach levels was not the scope of our investigation but we now mention in the discussion several studies, which have analyzed the effects of choline deprivation on this neurotransmitter levels in rodents. While long-term, dietary restriction of choline can affect extra- and intracellular sources of substrates required for Ach synthesis, and eventually limit Ach release in the hippocampus (Nakamura et al., 2001), this may vary in discrete regions of the brain and compensatory mechanisms exist. Among them upregulation of the expression of the high affinity choline transporter (Mellott et al., 2007) or of the acetylcholine synthesizing enzyme choline acetyltransferase (Batra et al., 2011).

  1. The same pattern was repeated in Figs. 3 and 4. The data do not support strong connections between CFD effects and the ASM knockout, because wt mice showed the same effects. 

We believe Figure 3 evidences two important points regarding the safety of CFD for the period of time used in our experiments (2 months). On one hand, CFD does not induce inflammation in the wt mice arguing against the concern that this diet would be harmful for the liver. On the other hand, CFD not only does not induce inflammation but reduces that already present in the liver of ASMko mice arguing in favor of a beneficial, although limited, effect in the disease phenotype.

Regarding Figure 4, we agree with the reviewer that CFD has almost no impact in the brain lipid composition of both wt and ASMko mice. We have now added new panels (E,H,K), which highlight the lack of significant changes in the relative abundance of SM, PC and PE species in both mouse genotypes.

  1. The data presented are difficult to understand at the metabolic circuit level, and need to have a clear physiological model for better presentation and easy understanding.

We mention in the discussion some metabolic circuits that may explain the CFD mild effects in the liver and lack of impact in the brain lipidomics in wt and ASMko mice (i.e. diminished CerS2 activity (lines 388-392), increased PEMT activity (lines 433-435), downregulation of the Kennedy pathway (Lines 427-433). Given the limited impact of CFD in the ASMko mice, and to avoid opening groundless expectations about its application to neurovisceral ASMD patients, we have removed along the manuscript (title and last sentences in abstract, introduction and discussion) the positive statements about its therapeutical potential. We have replaced them for sentences in which we highlight the limited efficacy of this approach. We hope the additions made in the text satisfy the reviewer request to clarify underlying metabolic defects and improve understanding. 

Comments on the Quality of English Language. Moderate editing

The English Language was revised by one of the authors who is native speaker. 

Reviewer 2 Report

Finding new approaches to treat the neuronopathic component in acid sphingomyelinase deficiency forms with brain involvement is a difficult challenge, and any innovative postulate should be worth a trial. That of a dietary choline deprivation, however, seems at first somewhat far reached, since choline deficiency has been associated to hepatic steatosis, and also learning disabilities or memory loss. On the other hand, a report (ref 19) has shown that ASMase deficiency ameliorates resistance to choline/methionine deficient- or high-fat diet-induced steatosis, with further data indicating that ASMase is required for autophagy-induced ER stress.

Based on these data, the present study aimed to investigate the effect of an 8-week choline-free diet (less toxic than the choline/methionine diet) both in wild-type and ASMko mice, aged 4.5 months at the time of analysis, the working hypothesis being to try to reduce sphingomyelin accumulation by limiting its production.

The study design is clear, analytical methods are fully adequate and well described, and the manuscript is globally well written.

A useful information relates to the comparative lipid composition (sphingomyelin and lysosphingomyelin, major glycerophospholipids, but not cholesterol) in liver and cerebellum in control and ASMko mice at the age of 4.5 months. This completes a very recent study from the same authors (Gaudioso et al Cell Death Dis 2023) devoted to brain and cultivated neurons. The advanced lipidomics methodology provides a precise quantification of the level of accumulation of the studied lipids at this age, as well as their respective fatty acid distribution. In ASMko mice, the fatty acid composition of sphingomyelin (SM) appears clearly altered, with an increase of the long-chain 16:0, 18:0 and a decrease of the very long-chain fatty acids 24:0 24:1 in the liver. In cerebellum, an increase of 16:0 and a decrease of 24:1 seem to be the most conspicuous changes. Modifications in the distribution of fatty acid species of phosphoglycerides appear less obvious.

Importantly, no overt negative effect of the diet was evidenced in the control mice regarding weight, liver to body weight, motor abilities, markers of inflammation either in liver or cerebellum, and number of Purkinje cells. In liver, the diet reduced sphingomyelin by 1.3-fold (and induced a change in fatty acid composition to a profile intermediate between that in untreated controls and ASMko mice).

However, the effect of the same 8-week choline-free diet in ASMko mice appeared minimal:

-        No change in the global concentration of sphingomyelin (and lysosphingomyelin) nor their fatty acid distribution, either in liver or cerebellum. 

-         No effect on preservation of Purkinje cells (but what is the status at initiation of diet?)

-          No effect on motor impairment

-          There was a minor effect on inflammation in liver and in microglia size in cerebellum.

The authors rightly discuss that although a downregulation of the Kennedy pathway was likely, the presence of methionine in the diet and the sustained formation of choline phosphoglycerides from ethanolamine phosphoglycerides could explain the failure to reduce sphingomyelin levels in the ASMko mice. So, overall, the results are essentially negative.

Main points  

1.     1. A main criticism is the too optimistic last sentence in the abstract, even though the results are rightly reported essentially negative in the preceding sentence. The phrasing should be altered to a more realistic view. Even lines 394-398 in the discussion do not take into consideration the extreme practical limitations of a possible extrapolation/ application to human patients with neuronopathic ASMD. This approach remains very theorethical.

2.      2. Study design: the authors should have better explained the timing of onset and duration of the diet in relation to the course of disease in the ASMko mouse. According to the brief clinical description in ref 21, these mice are early symptomatic at 6-8 (?) weeks of age, and indeed the choice of this age makes sense. This is an important point. Purkinje cells, however, might already be gone at this time (indicate, if known); an amelioration could then hardly be expected. Similarly, is dynamics of lipid accumulation known? Does a further sphingomyelin accumulation occur between 6 -8 weeks and 4.5 months in untreated ASMko mice?

3.   3. There is an ambiguity regarding exact timing for treatment of mice, and age at analysis:  In IJMS, Materials and Methods are at the end of the manuscript, which is generally perfectly all right. However, analytical results should not be reported without having more precisely defined the general design of the experiments in a very short but separate section at the beginning of the results part. Besides, this has resulted in confusing numbers. Currently, there is a discrepancy between what is written on line 71 and on lines 405 and 412. Line 71 indicates that the choline-free diet was initiated in 6-week-old mice but line 412 says 8-week-old mice. Both instances note that it was pursued for 8 weeks; however, neither of the total periods (14 or 16 weeks) fits with line 405 saying that the analyses were performed at 4.5 months of age. This must be clarified. The age at which the analyses were performed is particularly important in relation with the brain data shown in Fig. 5.

4.     4. The relative fatty acid distribution is easier to appreciate in the liver (from Fig 2 E,H,K schemes) than in cerebellum (Fig 4). In the latter, a similar presentation of the data should have been added (even though the sphingomyelin species distribution graph in untreated mice was used in the other paper).

5.       5.  Section 2.1: “Choline-free diet is not toxic and does not affect liver to body weight ratio in ASMko mice:  the authors should perhaps be more careful:  while no overt toxicity was apparent under the experimental conditions (after 8 weeks, no increase of the liver to body weight even in the wt mice),  it may be too far reached to conclude to a non toxicity – the diet was not pursued longer (this could have been done in some of the mice). and no comparative (before/after) histologic study of the liver was performed. Food consumption clearly declined in the ASMko mice fed the choline-free diet and the same trend is seen for wt mice, although it was not significant. The discussion (lines 313-324) could better indicate the limitations of the study.  

Minor points

1.       2.2: Choline-free diets… mainly by changing levels of sphingo- and glycerophospholipids with 34:1 fatty acids: should be modified to “… changing levels of sphingolipids and glycerophospholipids with…” 

2.       Figure 5C (rotarod experiment) should somehow stand out in a better manner. As of now, the information has no visibility.

3.       Inadequate referencing:

-          Ref 3 which is a short case report focusing on SMPD1 mutations is definitely not suitable to illustrate neuroinflammation and neurodegeneration.

-          Ref 6: among references from the group, the Ledesma et al J Neurochem 2011 review would be more appropriate here.

-          Line 114: Gaudioso et al 2023 in press: a paper in press can be cited, and this paper seems in fact to be published (very likely: Gaudioso et al Cell Death Dis. 2023;14:248).

4.       Correct the (very few) typos: Line 42: phosphocoline; line 145 coline-free; line 306: therfore; line 384: phosphocoline

Author Response

REVIEWER 2

We thank this reviewer for considering that the study design is clear, the analytical methods fully adequate and well described, the results rightly discussed and the manuscript well written. We are also grateful for his/her comments and suggestions that have been addressed as follows:

Main points 

  1. A main criticism is the too optimistic last sentence in the abstract, even though the results are rightly reported essentially negative in the preceding sentence. The phrasing should be altered to a more realistic view. Even lines 394-398 in the discussion do not take into consideration the extreme practical limitations of a possible extrapolation/ application to human patients with neuronopathic ASMD. This approach remains very theorethical.

We agree with the reviewer and have now changed the last sentences in the abstract and the discussion to offer a more realistic view of the therapeutic potential of choline dietary deprivation for neuronopathic ASMD. We have also removed the last sentence of the introduction, since it gave a too optimistic view of this strategy, and eliminated from the title the reference to the potential therapeutic approach.

  1. Study design: the authors should have better explained the timing of onset and duration of the diet in relation to the course of disease in the ASMko mouse. According to the brief clinical description in ref 21, these mice are early symptomatic at 6-8 (?) weeks of age, and indeed the choice of this age makes sense. This is an important point. Purkinje cells, however, might already be gone at this time (indicate, if known); an amelioration could then hardly be expected. Similarly, is dynamics of lipid accumulation known? Does a further sphingomyelin accumulation occur between 6 -8 weeks and 4.5 months in untreated ASMko mice?

We thank the reviewer for making this point that is indeed important to clarify. We now do it in lines 75-81. We started to fed wt and ASMko mice a control or choline-free diet at eight weeks of age. At this stage Purkinje cell degeneration is evident in the anterior, but not posterior, lobes of the cerebellum causing the first disease symptoms such as tremors and motor impairment in the ASMko mice (Macauley et al., 2008). The choline-free diet was extended for eight weeks, a period of time in which Purkinje cell death and SM accumulation progress in untreated ASMko mice (Horinouchi et al., 1995, Macauley et al., 2008).

  1. There is an ambiguity regarding exact timing for treatment of mice, and age at analysis: In IJMS, Materials and Methods are at the end of the manuscript, which is generally perfectly all right. However, analytical results should not be reported without having more precisely defined the general design of the experiments in a very short but separate section at the beginning of the results part. Besides, this has resulted in confusing numbers. Currently, there is a discrepancy between what is written on line 71 and on lines 405 and 412. Line 71 indicates that the choline-free diet was initiated in 6-week-old mice but line 412 says 8-week-old mice. Both instances note that it was pursued for 8 weeks; however, neither of the total periods (14 or 16 weeks) fits with line 405 saying that the analyses were performed at 4.5 months of age. This must be clarified. The age at which the analyses were performed is particularly important in relation with the brain data shown in Fig. 5.

We apologize for the confusion on the timing of treatment. We now clarify in all instances that it started at 8 weeks of age and lasted for another 8 weeks, thus finishing at 16 weeks of age.

  1. The relative fatty acid distribution is easier to appreciate in the liver (from Fig 2 E,H,K schemes) than in cerebellum (Fig 4). In the latter, a similar presentation of the data should have been added (even though the sphingomyelin species distribution graph in untreated mice was used in the other paper).

We now include a similar presentation of the data by adding new panels E, H and K in Figure 4 showing bars with the relative abundance of SM, PC and PE species, respectively, in the cerebellum of wt and ASMko mice treated with control or choline-free diet.

  1. Section 2.1: “Choline-free diet is not toxic and does not affect liver to body weight ratio in ASMko mice: the authors should perhaps be more careful:  while no overt toxicity was apparent under the experimental conditions (after 8 weeks, no increase of the liver to body weight even in the wt mice), it may be too far reached to conclude to a non toxicity – the diet was not pursued longer (this could have been done in some of the mice). and no comparative (before/after) histologic study of the liver was performed. Food consumption clearly declined in the ASMko mice fed the choline-free diet and the same trend is seen for wt mice, although it was not significant. The discussion (lines 313-324) could better indicate the limitations of the study.

The reviewer is right. We now stress the limitations of the study in lines 400-409 of the discussion.     

Minor points

  1. 2: Choline-free diets… mainly by changing levels of sphingo- and glycerophospholipids with 34:1 fatty acids: should be modified to “… changing levels of sphingolipids and glycerophospholipids with…”

Corrected

  1. Figure 5C (rotarod experiment) should somehow stand out in a better manner. As of now, the information has no visibility.

We have modified the graph in Figure 5C by specifying it reflects the Rotarod test and by indicating in the Y axis that we measured the time spent by the mice on the rotating rod.

  1. Inadequate referencing:

-          Ref 3 which is a short case report focusing on SMPD1 mutations is definitely not suitable to illustrate neuroinflammation and neurodegeneration.

-          Ref 6: among references from the group, the Ledesma et al J Neurochem 2011 review would be more appropriate here.

-          Line 114: Gaudioso et al 2023 in press: a paper in press can be cited, and this paper seems in fact to be published (very likely: Gaudioso et al Cell Death Dis. 2023;14:248).

We apologize for the inadequate referencing that has been corrected.

  1. Correct the (very few) typos: Line 42: phosphocoline; line 145 coline-free; line 306: therfore; line 384: phosphocoline

We thank the reviewer for the thorough revision. The typos have been corrected.

Round 2

Reviewer 1 Report

The revised manuscript addressed some of the points I raised; however, it still failed to address the key points regarding the CD and CFD-fed mice. Based on the metabolic cycles in Fig. 6, ASM deficiency should lead to SM accumulation, which was detected as (Fig. 2 and 4). On the other hand, CFD feeding was expected to decrease significantly total PC level and reduce the accumulated SM level, in comparison with the CD-fed mice (of both genotypes), which were not detected. The fact that such fundamentally important points being were refuted by the data presented in the current paper either suggests that the whole metabolic routes presented in Fig. 6 are incomplete, if not wrong, or indicates that the data contrasting the differences between CD- and CFD-fed mice were incorrectly obtained or analyzed. Either of these ruins the conceptual basis for the key hypothesis the authors wanted to advance. If the presented data were right and statistically sound, the authors should carefully analyze what went wrong with the metabolic pathways so far known, instead of presenting the self-contradicting data and model.

English is much better. 

Author Response

The revised manuscript addressed some of the points I raised; however, it still failed to address the key points regarding the CD and CFD-fed mice. Based on the metabolic cycles in Fig. 6, ASM deficiency should lead to SM accumulation, which was detected as (Fig. 2 and 4). On the other hand, CFD feeding was expected to decrease significantly total PC level and reduce the accumulated SM level, in comparison with the CD-fed mice (of both genotypes), which were not detected. The fact that such fundamentally important points being were refuted by the data presented in the current paper either suggests that the whole metabolic routes presented in Fig. 6 are incomplete, if not wrong, or indicates that the data contrasting the differences between CD- and CFD-fed mice were incorrectly obtained or analyzed. Either of these ruins the conceptual basis for the key hypothesis the authors wanted to advance. If the presented data were right and statistically sound, the authors should carefully analyze what went wrong with the metabolic pathways so far known, instead of presenting the self-contradicting data and model.

We respectfully disagree with this reviewer comment suggesting that the data were incorrectly obtained or analyzed. Lipidomic analysis was performed by an expert research unit (RUBAM) at the Institute of Advanced Chemistry of Catalonia (www.rubam.net), which is specialized in sphingolipid quantification and counts with state-of-the-art equipment. The clear differences we observed between wt and ASMko mice fed with CD (including increased levels of all SM species, LysoSM, dhSM, PE and PC) that have been previously reported by others, validate the data collection and analysis.

The reviewer is right that Figure 6 did not reflect the full complexity of the metabolic pathways for sphingolipid synthesis. The figure was a simplified view reflecting the two pathways that may explain the results obtained and are mentioned in the discussion section. Down regulation of the Kennedy pathway may account for the lack of effect of CFD on PC levels (discussed in lines 373-379). The presence of methionine in the diet, and consequent activity of PEMT, may compensate choline deprivation and keep unchanged PC and SM levels under CFD (discussed in lines 379-381). To satisfy this reviewer we have now modified the figure 6 to better reflect the complexity of the metabolic routes for sphingolipids and phospholipids.

Finally, we agree with the reviewer the results obtained do not sustain our initial hypothesis and discourage choline deprivation as a nutraceutical strategy to fight against ASMD. We believe that, although negative, this is an important message to communicate, especially for the ASMD community in which choline-free diet has been proposed as a suitable treatment. To make this point clear we have now added a sentence in the discussion (lines 340-341). Please see also the last sentence of the discussion.

Reviewer 2 Report

The working hypothesis – beneficial effect of an 8-week choline-deficient diet in ASM-ko mice as a potential therapeutic approach in acid sphingomyelinase deficiency (ASMD) – led to results which should be considered as essentially negative (to summarize the findings: there was no change in levels of sphingomyelin, lysosphingomyelin, PC or PE, Purkinje cells profile, or motor (rotarod) performance. The only positive effects were on macrophage size (not number) in liver and microglia size (not number) in cerebellum).

In their interpretation of the results in the original manuscript, the authors largely minimized this fact. Therefore, the title, as well as some sentences in the abstract and discussion were misleading. This was in large part corrected in the revised version, but a few statements could still be amended.

Title: The new title indeed better reflects the content of the manuscript. To help the reader, it would be best to indicate that the study was conducted in a mouse model:

“Modulation of dietary choline uptake in a mouse model of acid sphingomyelinase deficiency”

Abstract:

Remaining remark: Line 22: “The impact on sphingolipid levels was limited …” is still misleading, if the statement is addressing ASMko mice, as seems the case. Data (lines 126-136; 226-230) show that in the ASMko mice, there was NO impact on sphingolipid total levels, which is the most important observation, NOR significant changes in their fatty acid species relative abundance. On the other hand, in wt mice, there was for sphingomyelin an effect on both the total level, and species distribution. Here, since the authors only summarize effects on the ASMko mice, writing that there was no impact would be more correct.

Results:

Modifications on lines 73-80 and additional referencing regarding the ASMko mice are useful and respond to my critique.

Question: On line 103, why a hyphen after “sphingolipid-“? this is unclear as it seems to refer to sphingolipid species with 34:1 fatty acid, which is of course meaningless (the text in lines 126-136 is clear).

Amendments in the section “brain lipid composition” (text and Figure 4) and the rest of the results section also answer my remarks.

Discussion:

Lines 406-413 and related references 50-52 constitute a useful addition.

Remaining remark: Last sentence (lines 428-431) still sounds a bit too optimistic in the view of this referee. From the current results, apart from the difficulty to administrate a choline-free diet to human patients, there is no indication that this treatment alone would have a meaningful impact.

References:

Added references are pertinent. I do not understand for which reason reference 4 was kept, but this is a detail.

Remaining typos foundThis is:

Line 75: “we started to fed wild type and ASMko mice”:  must be: “we started to feed…”.

Line 297: ammelioration instead of amelioration

Line 417: phoshocoline instead of phosphocholine

Line 428: “a extrapolation” should be “an extrapolation”.

Author Response

We thank this reviewer for considering that the queries were in large part addressed in the revised version and for indicating a few statements that could still be amended. We have done so as follows:

Title: The new title indeed better reflects the content of the manuscript. To help the reader, it would be best to indicate that the study was conducted in a mouse model: “Modulation of dietary choline uptake in a mouse model of acid sphingomyelinase deficiency”

We have corrected the title accordingly

Abstract:

Remaining remark: Line 22: “The impact on sphingolipid levels was limited …” is still misleading, if the statement is addressing ASMko mice, as seems the case. Data (lines 126-136; 226-230) show that in the ASMko mice, there was NO impact on sphingolipid total levels, which is the most important observation, NOR significant changes in their fatty acid species relative abundance. On the other hand, in wt mice, there was for sphingomyelin an effect on both the total level, and species distribution. Here, since the authors only summarize effects on the ASMko mice, writing that there was no impact would be more correct.

We now clearly state in the abstract that there was no significant impact on sphingolipid levels

Results:

Question: On line 103, why a hyphen after “sphingolipid-“? this is unclear as it seems to refer to sphingolipid species with 34:1 fatty acid, which is of course meaningless (the text in lines 126-136 is clear).

We have eliminated hyphens in all instances

Discussion:

Remaining remark: Last sentence (lines 428-431) still sounds a bit too optimistic in the view of this referee. From the current results, apart from the difficulty to administrate a choline-free diet to human patients, there is no indication that this treatment alone would have a meaningful impact.

We have deleted the positive statement in the sentence.

References:

Added references are pertinent. I do not understand for which reason reference 4 was kept, but this is a detail.

Reference 4 refers to the benefits of intravenous infusion of the recombinant ASM in ASMD patients. We believe it is appropriate for the text.

Remaining typos found: We apologize for the wrong typos, which have been corrected

Round 3

Reviewer 1 Report

The revision still did not change the general conclusion that the data presented do not support the key hypothesis that "choline deprivation could reduce SM production and have beneficial effects in ASMD". Further, the data do not support directly the conclusion on "..  a modest potential of this nutritional strategy to assist in the management of neurovisceral ASMD patients". The reduction in inflammation in liver in Fig. 3B is small, compared to the level in WT mice. The change in microglia size is not a direct marker for reduced inflammation in the brain, either (Fig. 5F). Other direct molecular markers for inflammation should be used in a direct quantitative fashion, instead.  

I thus maintain the standing that generally negative data not supporting the main conclusions and the key hypotheses undermine the basis for publication. 

The organization of the paper should be changed because of the general negative data not supporting the key hypothesis. 

Author Response

To address the remaining concerns of Reviewer 1 and according to the academic editor's comments we have done the following modifications in the text (also indicated in the manuscript copy with tracked changes).

-In the abstract, we have replaced the last sentence: " … indicating a modest potential of this nutritional strategy to assist in the management of neurovisceral ASMD patients” for “… arguing against the potential of this nutritional strategy to assist in the management of neurovisceral ASMD patients”.

- To be more precise describing our observations we have deleted the reference to inflammation in the abstract and in lines (63-65; 203; 207; 305-306; 425-427) and specify that the effects we have observed concern the size of macrophages and microglia.

-We believe the changes made in the abstract contribute to alter the organization of the paper to further underscore that the general conclusion does not support the initial hypothesis of our work.